# Evaluation of a Condition Monitoring Algorithm for Early Bearing Fault Detection

**DOI:** 10.3390/s24072138

**Published:** 2024-03-27

**Authors:** Hannes Gruber, Anna Fuchs, Michael Bader

**Affiliations:** 1AVL List GmbH, Hans-List-Platz 1, 8020 Graz, Austria; anna.fuchs@avl.com; 2IME (Institute of Machine Components), TU Graz (Graz University of Technology), 8010 Graz, Austria; michael.bader@tugraz.at

**Keywords:** condition monitoring, bearing failure, condition indicator, roller bearing, driveline testing, engine testing, test bed condition monitoring, damage detection

## Abstract

Roller bearings are critical components in various mechanical systems, and the timely detection of potential failures is essential for preventing costly downtimes and avoiding substantial machinery breakdown. This research focuses on finding and verifying a robust method that can detect failures early, without creating false positive failure states. Therefore, this paper introduces a novel algorithm for the early detection of roller bearing failures, particularly tailored to high-precision bearings and automotive test bed systems. The featured method (AFI—Advanced Failure Indicator) utilizes the Fast Fourier Transform (FFT) of wideband accelerometers to calculate the spectral content of vibration signals emitted by roller bearings. By calculating the frequency bands and tracking the movement of these bands within the spectra, the method provides an indicator of the machinery’s health, mainly focusing on the early stages of bearing failure. The calculated channel can be used as a trend indicator, enabling the method to identify subtle deviations associated with impending failures. The AFI algorithm incorporates a non-static limit through moving average calculations and volatility analysis methods to determine critical changes in the signal. This thresholding mechanism ensures the algorithm’s responsiveness to variations in operating conditions and environmental factors, contributing to its robustness in diverse industrial settings. Further refinement was achieved through an outlier detection filter, which reduces false positives and enhances the algorithm’s accuracy in identifying genuine deviations from the normal operational state. To benchmark the developed algorithm, it was compared with three industry-standard algorithms: VRMS calculations per ISO 10813-3, Mean Absolute Value of Extremums (MAVE), and Envelope Frequency Band (EFB). This comparative analysis aimed to evaluate the efficacy of the novel algorithm against the established methods in the field, providing valuable insights into its potential advantages and limitations. In summary, this paper presents an innovative algorithm for the early detection of roller bearing failures, leveraging FFT-based spectral analysis, trend monitoring, adaptive thresholding, and outlier detection. Its ability to confirm the first failure state underscores the algorithm’s effectiveness.

## 1. Introduction

Condition monitoring (CM) is a pivotal strategy in industrial maintenance, promising substantial cost savings by preemptively identifying equipment damage before it escalates into costly failures. This transition from reactive to planned maintenance has significantly reduced downtime costs, with estimates that the actual cost of downtime in manufacturing is up to USD 260,000 an hour [1]. Forecasts show that the overall market value of the predictive maintenance sector is expected to grow exponentially from USD 5.7 bn in 2021 up to USD 64 bn in 2030 [2].

The application of CM extends beyond traditional high-value equipment in various industries as advancements in sensor technology and data processing have democratized its adoption across diverse industrial sectors. This paradigm shift is particularly evident in industries like electrified automotive drivelines [3] and railway applications [4]. However, the realization of CM’s whole potential is based on identifying reliable condition indicators (CIs) that are capable of accurately discerning incipient failures [5]. In particular, automotive applications deal with a wide range of uncertainties, e.g., a wide variety of operating conditions like temperature [6], road surface conditions [7], speed, and torque gradients [8] as well as driver input errors and highly dynamic disturbances like wideband torsional excitations through blocking wheels [9,10] or controller-based safety measures (Electronic Stability Programs (ESPs) and others) [11], which makes a limit monitoring approach using steady state limits unable to detect failures of mechanical components at early stages [12]. Modern development tools for automotive powertrains must reproduce these disturbances and withstand these conditions for long testing periods [13]. Detecting machine conditions is necessary to provide a constant platform for comparable tests and an indicator for predictive maintenance, where an unexpected shutdown of the test bed results in a loss of data quality and valuable development time [12,13].

This paper introduces a novel failure indicator tailored for roller bearing monitoring applications developed for detecting failures on high-speed precision roller bearing setups in test beds for the automotive industry. As an independent basis for evaluating the indicator algorithm, it was applied to the publicly available NASA Bearing Dataset [14] using three indicators: VRMS calculations per ISO 10813-3 [15], Mean Absolute Value of Extremums (MAVE), and Equivalent Frequency Band (EFB) [16]. An emphasis was placed on sensitivity to parameter tuning and failure to identify and minimize false alarms [17].

VRMS calculations, as described in the above-mentioned ISO Standard, are designed to quantify vibrations in industrial machines with a net electrical power of more than 15 kW and rotational speeds between 120 and 15,000 RPM, providing an easy-to-calculate, time-domain-based calculation for failure detection by calculating the vibration velocity through the integration of an acceleration signal, bandpass filtering between 10 and 1000 Hz, and a final RMS calculation [18]. Due to the bandpass filtering, the resulting failure indicator focuses on operational corresponding fault frequencies, which tend to occur at a late failure stage.

The MAVE method, characterized by removing noise, extracting the local extremum, and calculating the MAVE RMS values, is performed in the time domain [19], making it a reasonable choice for limited computational power and online condition monitoring. With the increased computational capabilities of condition monitoring hardware, frequency-domain-based calculations can be performed online [20], enabling a deeper insight into the vibration data compared to pure time-domain-based approaches.

Unlike the described approaches centered on acoustic emissions or vibration analysis in the time domain, the proposed AFI method evaluates the complete bandwidth of the spectrum, encompassing signals beyond mere vibrations [21,22]. The technique eliminates transmission path effects by correlating the frequency band amplitude sums to an unfiltered total baseline, enhancing the diagnostic accuracy.

For limit monitoring, Schagerl [12] proposed methods that track or predict the trend of the indicator value, as described in Figure 1, by either using machine learning or, in the case of unknown operating conditions, to track the measured narrow-band signals using statistical methods.

Based on these findings, a thresholding statistical method was used for dynamic limit monitoring based on statistical band tracking comparable to the Bollinger Bands in financial markets [23]. This method calculates a moving average of the historic indicator values, and the mean deviation of the historical values is used to calculate the bandwidth [24].

This paper presents a novel approach to bearing monitoring, offering a comprehensive evaluation of early failure detection algorithms for condition monitoring applications. The proposed method’s holistic spectrum analysis and practical implementation feasibility position it as a promising tool for enhancing equipment reliability and operational efficiency across various industrial sectors.

## 2. Methodology

### 2.1. Bearing Failure Detection in Vibration Data

To gain insight into the signal changes over the bearing fault stages, it is necessary to understand the mechanisms of typical bearing faults. Ball or race defects can be classified into 4 stages, where 1 is the lowest severity. Stage 4 defects lead to a total loss of function, which is characterized there being no stable operating temperature within the bearing manufacturer’s specification. It is likely that other components in the system (e.g., sealings, gears, couplings) are damaged as well. Figure 2 shows an interpretation of the bearing fault stages [17].

Each stage is defined by a characteristic frequency spectra pattern, as shown in Figure 3. Typically, the smaller the defect, the higher the corresponding frequency. This can be explained in the example of crack propagation. A small crack leads to a high eigenfrequency of the surrounding material, whereas a larger crack zone resonates at a lower frequency [25].

More severe bearing faults—stages 3 and 4—show lower frequency phenomena based on order-correlated phenomena like the ball passing frequency on the outer race (BPFO), the ball passing frequency on the inner race (BPFI), the ball spin frequency (BSF), and the fundamental train frequency (FTF) that are related to the movement of the cage rather than on uncorrelated wideband spectral phenomena [26].

The absence of all high-frequency lines characterizes an undamaged bearing frequency spectrum, and only order-based ground levels of system-immanent lines are present. Examples of such frequencies are 1st-order frequencies, caused by imbalance, and 2nd-order frequencies, caused by a Cardan joint error [27], which indicate looseness or a misalignment [28].

Stage 1 shows very high-frequency content in the Spike Energy/ultrasonic region; the system-immanent frequencies are unchanged at this stage. Zone 2 shows no bearing defects, and Zone 3 shows no bearing natural frequencies. These frequencies correspond to the eigenfrequencies of all structures in the bearing. Typically, these defects are not visible by physical inspection [29].

Stage 2 shows increased frequency values compared to those in Stage 1 and frequencies in the range of the bearing natural eigenfrequencies. Bearing defect frequencies are not found. The system-immanent frequencies remain unchanged. A physical inspection would show minor defects [29].

Stage 3 is characterized by the order-based frequencies, as explained in Figure 3. Zone 3 and 4 frequencies are elevated. A physical inspection would show larger areas of defects without loose particles [29].

Stage 4 shows an increased level of system-immanent, order-based frequencies, increased modulated frequencies (sidebands), as well as increased amplitudes in bearing natural frequencies [29].

Figure 4 outlines the characteristic bearing defect frequencies with the corresponding formulas. D is the diameter between the centers of rotation of two opposite rolling elements. d is the rolling element diameter. RPM represents the rotational speed in revolutions per minute and a is the bearing’s contact angle.

Based on the data, the algorithm was developed to detect failures before stage 3, which requires high-frequency data acquisition (a sampling rate higher than 5 kHz) [30]. The nature of the targeted application has high robustness against false positive failure detections and robustness against highly dynamic operating conditions. Therefore, robust outlier or operating-point detection with steady-state detection is required.

### 2.2. Algorithm Architecture

The following section outlines the algorithm’s architecture and the methods used to validate the proposed advanced failure indicator (AFI) to detect roller bearing failures early. The AFI algorithm by itself is structured as follows:Pre-processing of raw acceleration signals (filtering);Calculation of advanced failure indicators (AFI) in different frequency bands;Dynamic thresholding;Definition of outlier strategy;Tuning of hyper-parameters depending on specific bearing configuration;First time of failure detection (FTFD) or first time of prediction (FTP) calculation and plotting.

Figure 5 gives an overview of the calculation procedure of the four algorithms (AFI + 3 benchmarking methods). After signal conditioning of the respective raw data, the calculation rules are applied to the data. The resulting indicators are then compared against threshold levels, as defined by each method. If the threshold is violated and confirmed, the calculation sets an FTP or FTFD flag (used synonymously) for the corresponding method.

Due to its importance in practical applications, the VRMS algorithm is explained in further detail. Figure 6 outlines the procedure of the VRMS calculation following ISO 10816.

The algorithm is structured as follows:The raw acceleration signal is split up into 50 ms blocks;Low and high pass filtering (10 and 1000 Hz);Integration of the filtered signal;RMS calculation;Limit violation detection;Limit delay to increase robustness against dynamic operating conditions.

Due to the low-pass filtering, all higher-frequency content is removed from the signal, which limits the ability to detect early stages of failure [26].

The AFI calculation is described in the following subsections, which provide more details on the AFI (x), statistical band limit monitoring band (x), limit monitoring, and outlier detection blocks in Figure 5.

#### 2.2.1. Signal Conditioning and Feature Extraction

The first step involves acquiring high-frequency (larger than 20 kHz) vibration signals from roller bearings using acceleration sensors installed on the machinery. These raw vibration signals are preprocessed to remove noise and artifacts.

Subsequently, the Fast Fourier Transform (FFT) is applied to convert the time-domain vibration signals into the frequency domain, as shown in Figure 7. Important spectral features are extracted by decomposing the signals into frequency components, providing insights into the vibrational characteristics associated with bearing health for expert analysis. The AFI algorithm, however, does not take advantage of individual amplitudes in the spectra.

#### 2.2.2. Trend Monitoring (Zoning) Relative Indicator Evaluation and Adaptive Thresholding

The AFI utilizes trend monitoring of frequency band zones to track the behavior of spectral changes over time. A particular emphasis is placed on higher frequency components associated with early bearing fault stages. The filter zones are defined as follows:

Zone 0: unfiltered—used for normalization of the indicator value.

Zones 1–4: ascending frequency bins—Figure 8 shows example values for the frequency ranges.

The FFT spectral line amplitudes within the zone are summed up to create a total zone amplitude sum. To create the AFI, the individual amplitude sums for Zones 1, 2, 3, and 4 are divided by the total zone amplitude sum to obtained normalized AFIs 1, 2, 3, and 4, as outlined in Figure 9. These AFI channels are used for adaptive thresholding.

For thresholding, a moving average calculation establishes a baseline trend, while the volatility analysis method adaptively adjusts the detection thresholds based on changing operating conditions. The calculation procedure is outlined in Figure 10. This adaptive thresholding mechanism enhances the algorithm’s sensitivity to subtle deviations that are indicative of early failure stages while minimizing false alarms.

#### 2.2.3. Outlier Detection and Confirmation of Failure State

Figure 11 shows the trend monitoring procedure via the dynamic band limit, where a potential violation is evaluated. A limit violation is detected if an AFI zone sum value crosses the upper or lower band. An outlier detection filter is applied to confirm a failure state. This filter distinguishes genuine deviations associated with bearing failures from random fluctuations or noise in the data by requiring that at least two other zones and two consecutive, independent AFI calculations can confirm the violation. After this confirmation, the algorithm ensures the presence of a potential failure state in the roller bearing.

#### 2.2.4. Validation and Benchmarking

To validate the effectiveness of the AFI, a comparative analysis was conducted against industry-standard algorithms: VRMS calculations per ISO 10813-3 [16], Mean Absolute Value of Extremums (MAVE), and the Equivalent Frequency Band (EFB) [16] method. Performance metrics such as accuracy, sensitivity, and false positive rate were evaluated using benchmark datasets comprising real-world bearing failure scenarios. These comparisons provide insights into the AFI’s efficacy relative to established techniques, highlighting its potential for industrial application.

### 2.3. Benchmarking Data—The NASA Bearing Dataset

The publicly available NASA IMS Dataset, generated by the NSF I/UCR Center for Intelligent Maintenance Systems (IMS—www.imscenter.net) was chosen for this analysis due to its constant operating parameters. Further tests under dynamic operating conditions were performed, but the data cannot be reviewed publicly for confidentiality reasons.

The test rig setup consisted of four bearings installed on a shaft rotating at 2000 RPM, driven by an AC motor via a belt drive. As described in Figure 12, the bearings were subjected to a radial load of 26.5 kN. All failures occurred after the designed lifetime of 100 million revolutions.


**Metadata of the used sets:**


**Bearing Type:** ZA-2115 double row bearings, Rexnord;**Accelerometer:** PCB 353B33 High Sensitivity ICP mounted onto bearing housing; **Datasets for this analysis:** Datasets 2 and 3, with a single sensor mounted per bearing;**Recording strategy:** One measurement file every 10 min;**Sampling rate:** 20 kHz;**Description of Dataset 2:** Outer race failure after 164 h of runtime in Bearing No. 1;**Description of Dataset 3:** Outer race failure after 30 days and 21 h in Bearing No. 3;

## 3. Results

### 3.1. Descriptive Statistics

Before starting the evaluation, all datasets were analyzed for their data integrity. Therefore, the min, max, and mean values of the raw entries in each file were extracted from the file and plotted (Figure 13, Figure 14, Figure 15 and Figure 16).

The noise levels on the files were within a small range and constant over the entire dataset. Based on the data extracted from this descriptive analysis, all sensors were operative and set up correctly. There was no indication of clipping or other errors in the measurement chain.

### 3.2. VRMS and AFI Data

The vibration velocity calculation was used as a reference indicator. This method requires little computing power and is the reference for the ISO Standard 10816-3. It cleans the raw data with a DC filter and a 1000 Hz low-pass filter. Afterward, the data are integrated to calculate the vibration velocity. Figure 17 shows the bode plot of the VRMS method, where it is evident that all effects above 1000 Hz were effectively neglected.

The limit for this method is defined by ISO 10816-3, which classifies the maximum values. The permissible vibration velocity depends on the drive motor’s use case and electrical power supply. Figure 18 displays the table with the marked limit value for this calculation.

Figure 19 shows the results of the RMS calculations for Dataset No. 2. The dotted red line indicates the FTFD at Bearing No. 1 at a very late experimental stage. All other sensor positions did not identify a failure.

Figure 20 shows the results of the RMS calculations for Dataset No. 3. The dotted red line in the diagram indicates the FTFD at Bearing No. 3 at a very late experimental stage. All other sensor positions did not identify a failure.

Figure 21 introduces the graphical representation of one AFI zone. The unitless indicator level is plotted over the total measurement time of the dataset, which is translated into testing time at a later stage. The vertical blue lines indicate a band violation in that plotted zone. The vertical red line shows the first time that all four frequency zones identified limit violations simultaneously, representing the FTFD.

The calculation of the FTFD detection is based on the following hyper-parameters:-High-pass filter:○Filter frequency: 10 Hz;○Order: 6;○Characteristic: Butterworth.-Anti-aliasing filter:○Order: 10;○Characteristic: Butterworth.-FFT:○Lines: 8192;○Window function: Hamming;○Amplitude evaluation.-Limit band:○Bandwidth: 3 σ;○Standard deviation calc. length: Last 1440 AFI values;○Simple moving average length: Last 1440 AFI values.-False positive filtering:○Simultaneously creating limit violations in at least three zones with three consecutive identical outcomes.

The following charts (Figure 22, Figure 23, Figure 24, Figure 25, Figure 26, Figure 27, Figure 28 and Figure 29) give insights into the AFI results. Each figure represents one bearing and consists of 4 diagrams, each of them representing one AFI frequency zone, as explained in Section 2. The upper left zone represents the lowest frequency content, and the lower correct zone represents the one with the highest. The band indicates the zone-specific limit, which is used for limit detection, and the red dotted line shows the confirmed first prediction time (FTP) or first time of failure detection (FTFD).

**Figure 22 sensors-24-02138-f022:**
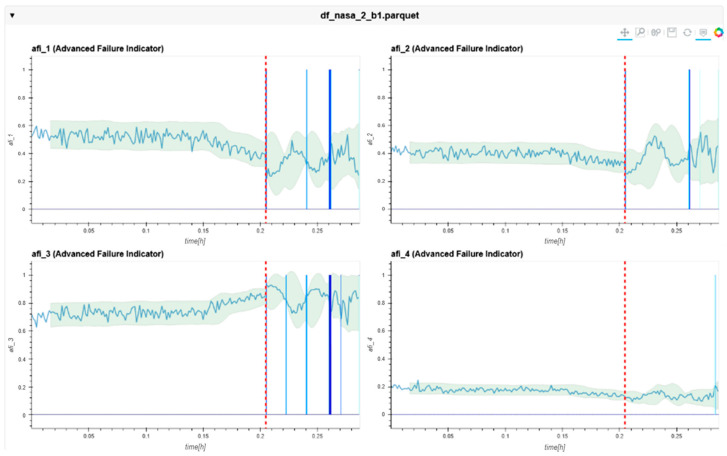
AFI plots for Dataset No. 2, Bearing No. 1. AFI [-] plotted vs. measurement time in hours.

For Dataset No. 2, Bearing No. 1, the FTFD was identified at 120 h into the experiment.

**Figure 23 sensors-24-02138-f023:**
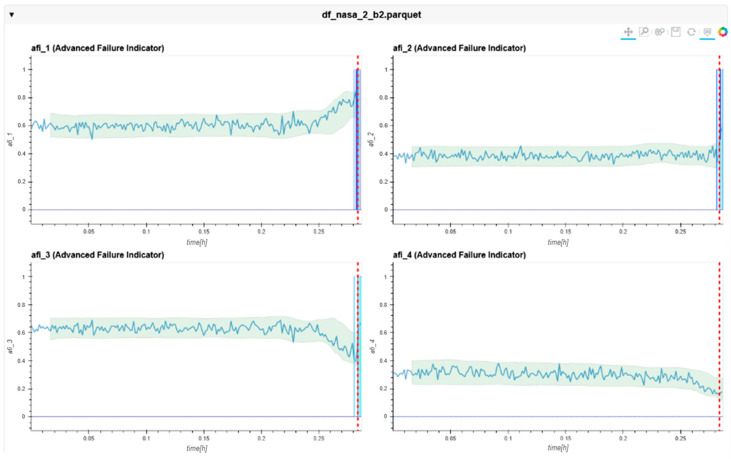
AFI plots for Dataset No. 2, Bearing No. 2. AFI [-] plotted vs. measurement time in hours.

For Dataset No. 2, Bearing No. 2, the FTFD was not identified until the end of the experiment.

**Figure 24 sensors-24-02138-f024:**
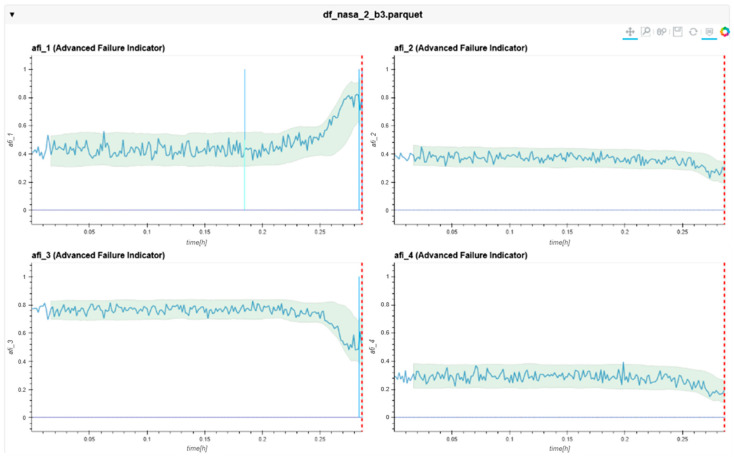
AFI plots for Dataset No. 2, Bearing No. 3. AFI [-] plotted vs. measurement time in hours.

For Dataset No. 2, Bearing No. 3, the FTFD was not identified until the end of the experiment.

**Figure 25 sensors-24-02138-f025:**
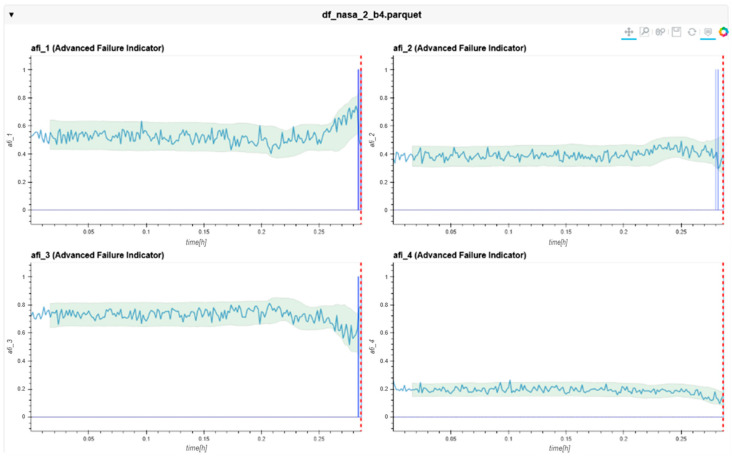
AFI plots for Dataset No. 2, Bearing No. 4. AFI [-] plotted vs. measurement time in hours.

The overall behavior of the AFI method showed a solid detection of the correct location at an earlier time than the RMS method.

Dataset No. 3 was calculated with the same hyper-parameter set as Dataset No. 2. Here, Bearing No. 1 showed no FTDF until the end of the experiment.

**Figure 26 sensors-24-02138-f026:**
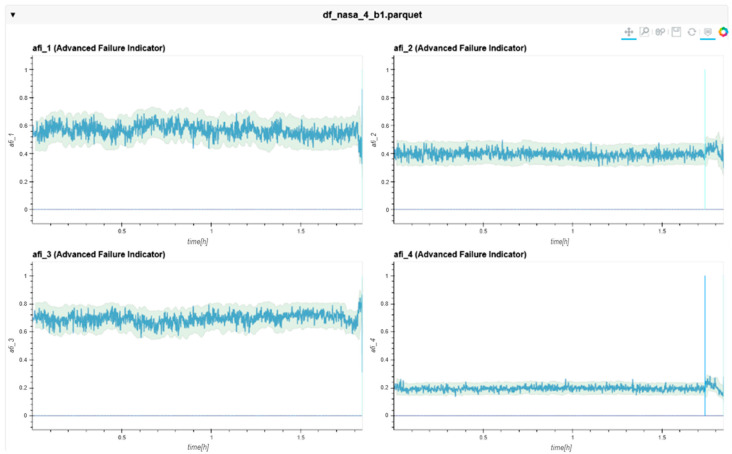
AFI plots for Dataset No. 3, Bearing No. 1. AFI [-] plotted vs. measurement time in hours.

**Figure 27 sensors-24-02138-f027:**
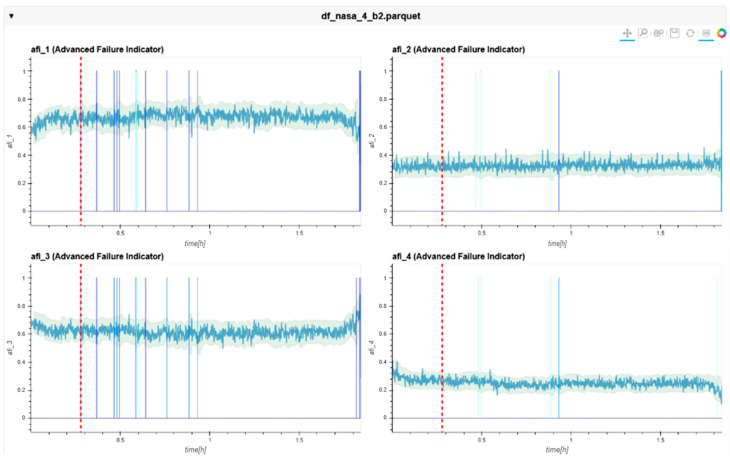
AFI plots for Dataset No. 3, Bearing No. 2. AFI [-] plotted vs. measurement time in hours.

For Dataset No. 3, Bearing No. 2, the FTFD was incorrectly identified at 168 h into the experiment.

**Figure 28 sensors-24-02138-f028:**
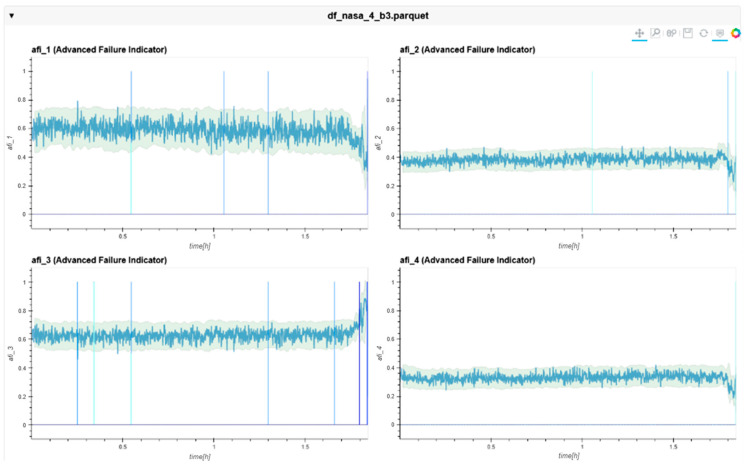
AFI plots Dataset No. 3, Bearing No. 3. AFI [-] plotted over measurement time in hours.

For Dataset No. 3, Bearing No. 3, the FTFD was not identified until the end of the experiment. Bearing No. 3 showed an outer race defect at the end of the test. Therefore, the AFI was not able to detect this failure correctly. The AFI method did not flag an FTFD because the limit band with its given parameters was not violated. Independent frequency zones show several violations within the zones (blue vertical lines). However, no timestamp was found, which satisfied the failure condition.

**Figure 29 sensors-24-02138-f029:**
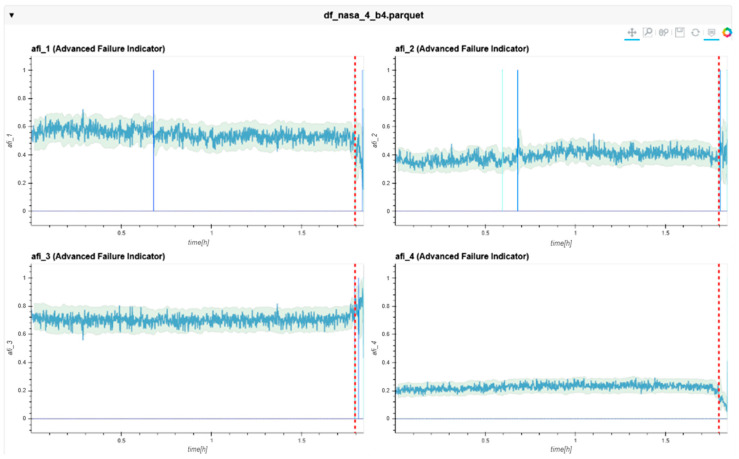
AFI plots for Dataset No. 3, Bearing No. 4. AFI [-] plotted vs. measurement time in hours.

For Dataset No. 3, Bearing No. 4, the FTFD was incorrectly identified at 744 h into the experiment.

### 3.3. Benchmarking Analysis—Validation and Robustness Comparison

To quantify the quality of the various methods, the main questions were:-Did the method detect the failure?-Did the method detect the failure earlier than the VRMS?-Did the method detect the failure between 60% and 80% of the absolute lifetime?-Did the method predict the correct bearing?-Did the method predict a failure too early (more than 40% remaining lifetime)?-Did the method predict a non-defect bearing (false positive)?

Figure 30 shows all FTFD results from Section 3.2 based on their sensor location and dataset to answer these questions.

All the methods detected failures for the Dataset No. 2, Sensor 1 (Bearing No. 1). As expected, the lowest frequency method, (V)RMS, reliably predicted the failure at a very late stage of the experiment, with only 2% of the remaining lifetime left. The EFB method detected failures in all bearings at various timesteps; the failing bearing was faulty at approximately 53 % of its lifetime, which is considered too early for detection. MAVE and AFI reliably detected the failure at 120 h, corresponding to 71 % of the absolute experiment duration.

The failure in Dataset No. 3 was, throughout all methods, more challenging to detect than that in Dataset No. 2. Throughout the overall runtime of 753 h, only the RMS method could detect the failure at the correct position. The failure was detected at 99% relative to the first time of failure detection, leaving only 1% (less than 10 h) lifetime stranded until the end of the experiment. MAVE failed to detect an error in time, and EFB saw an early failure at Sensor 4 (Bearing No. 4). AFI predicted a premature failure at Sensor 2 (Bearing No. 2) and a late failure detection at Bearing No. 4.

By clustering the data in Figure 30 and Figure 31, a ranking was made to quantify the most capable method and combination of the methods. As described in Figure 32, the velocity RMS method detected both failures reliably but at a timestamp that was reactive rather than predictive, which makes it a perfect shutdown criterion for monitoring—in Dataset No. 2, EFB, MAVE, and AFI detected the failure earlier than RMS. EFB predicted the failure at a very early stage at the correct location and, at a later stage, wrongly detected errors at all other sensors. MAVE predicted the failure simultaneously with AFI but with false positives at Sensor 2 and Sensor 4 at later timestamps.

As a prediction method, the AFI method performed flawlessly for Dataset No. 2, with a few shortcomings in Dataset No. 3, as shown in Figure 33. Since no other method could detect the failure at the correct position, the remaining KPIs were focused on the pure ability to detect a failure in the system, where both AFI and EFB produced false positive detections.

## 4. Summary, Limitations, and Challenges

This study supports the hypothesis that the AFI indicator can identify bearing failures earlier than the RMS method and is more robust than EFB and MAVE. The innovative approach of using normalized zone tracking and dynamic band limit monitoring leads to several benefits, such as a simple set-up procedure, the absence of a manual teach-in procedure, and efficient data reduction. However, further research is required to increase its stability for a dedicated industrial application (e.g., automotive test beds).

The combination of RMS and AFI seems to be the most promising way to obtain a reliable setup to detect bearing failures. AFI acts as a predictive indicator, ideally warning at an earlier stage than RMS, with a likelihood of reacting in the wrong position. RMS acts as a hard shutdown criterion for which the system must be analyzed and repaired. The hardware chosen for the desired application can simultaneously handle the complex AFI calculation, the data storage handling, and the ISO 10816 calculation. Therefore, both methods were used for further testing to quantify the effectiveness of the setup for the application at automotive engine test beds.

The datasets used for these calculations were produced at a constant speed and load, a test case in the automotive industry that is not necessarily repeated regularly. After this analysis, an operating class and steady-state detection were implemented to compensate for the non-stationary operating conditions. Meanwhile, the developed AFI algorithm was implemented and tested in real-world industrial environments, including customer test bed setups for race car development. Considerations were given to practical aspects such as computational efficiency, real-time processing requirements, and integration with existing condition monitoring systems. Feedback from industrial partners and end-users were solicited to refine the algorithm and ensure its practical usability in diverse operational applications.

The operating class detection clusters the relevant testing parameters (e.g., speed/torque/temperature) into a matrix defined by the test bed’s operating range to classify the stored results and make them comparable. The steady-state detection calculates the derivate of the testing parameters and ensures quasi-static conditions for the stored data. The data with high speed/torque/temperature gradients are considered invalid and not stored for further analysis. The absolute shutdown based on VRMS is permanently active and not triggered by the steady state and operating class detection.

While the AFI shows promise for the early detection of bearing failures, it is essential to acknowledge the potential limitations and areas for improvement. Factors such as sensor placement, environmental noise, and variability in operating conditions may influence the algorithm’s performance. Future research should involve refining the algorithm, expanding its applicability to different bearing types and machinery, and exploring advanced machine learning techniques for enhanced predictive capabilities. Another limitation of the AFI can be found in the summation of the FFT zones, which prevents the detailed identification of the source of the increased vibration. For example, an increase in the signal due to an early bearing failure is indistinguishable from a wearing spline coupling. To distinguish between such effects, an order-based analysis of FFT is required. Further work is needed to improve the reliability of this method. The tuning of the limit parameters is a promising approach to achieve this, as well as improved band calculation methods.

The complexity of modern automotive test beds requires extensive limit monitoring to ensure safe operation, and the AFI method can contribute to the safety of mechanical components. Parameter tuning is necessary for each application to achieve the target of predictive maintenance and keep the FTFD within acceptable ranges. Detecting a failure too early results in unnecessary stock costs for spare parts, while detecting a failure too late leads to expensive standstill time.

## Figures and Tables

**Figure 1 sensors-24-02138-f001:**
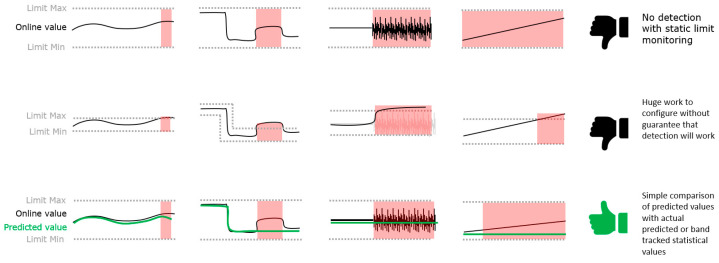
Limit monitoring techniques and limitations, as published by Schagerl [12].

**Figure 2 sensors-24-02138-f002:**
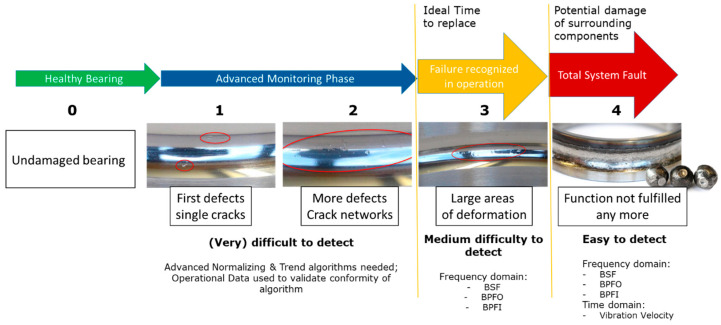
Bearing fault stages, race defects, and difficulty in detection [17].

**Figure 3 sensors-24-02138-f003:**
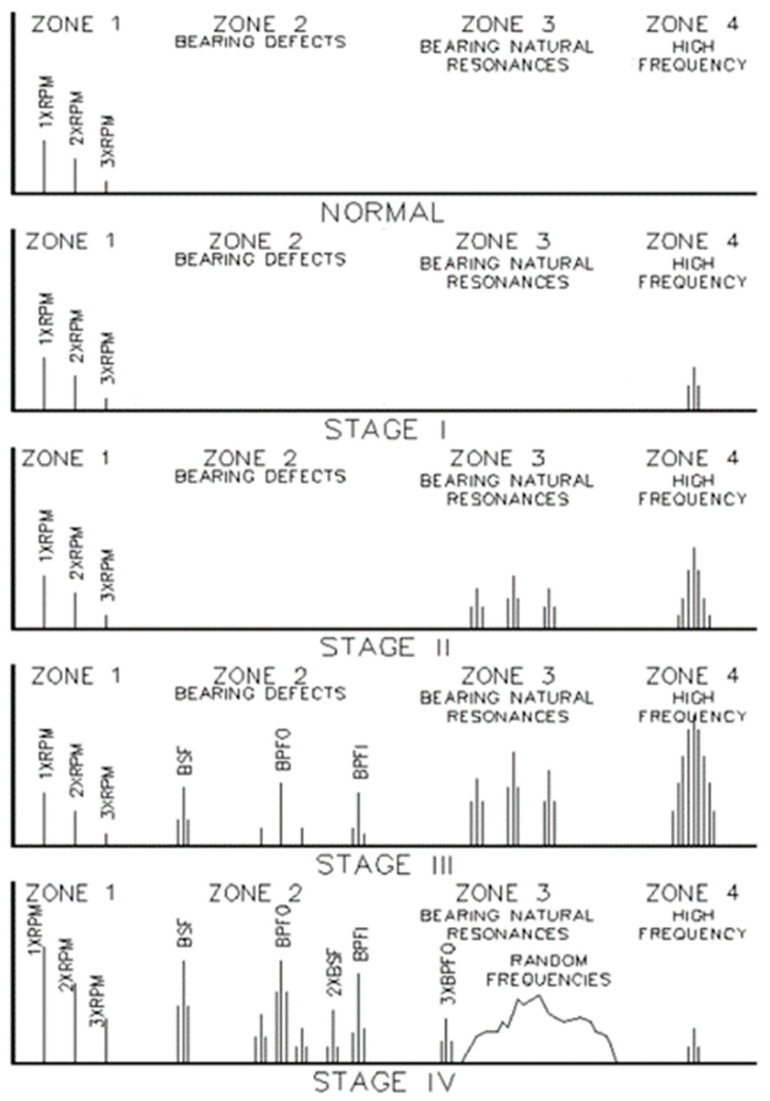
Bearing fault stages and corresponding spectral phenomena [26].

**Figure 4 sensors-24-02138-f004:**
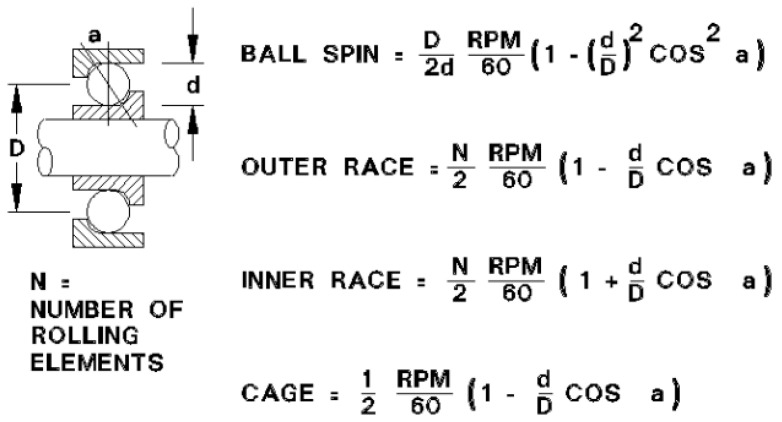
Order-based bearing fault frequencies [29].

**Figure 5 sensors-24-02138-f005:**
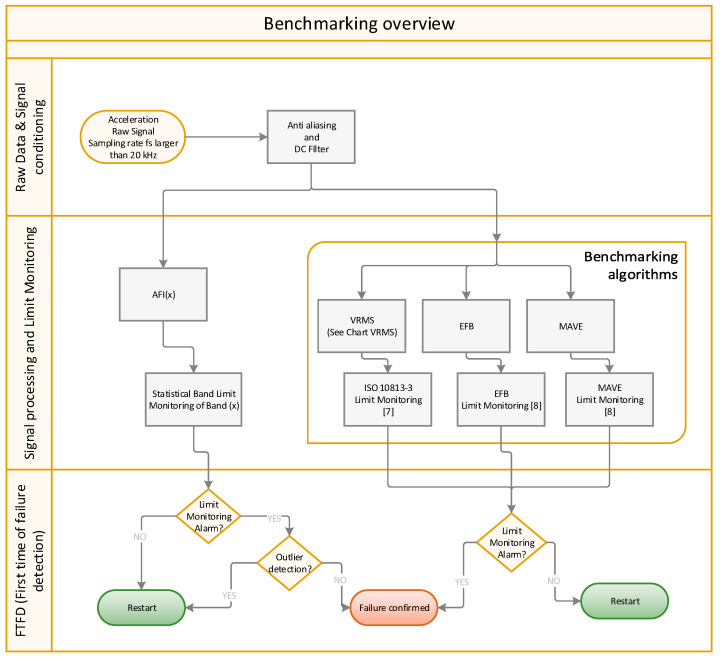
Benchmarking overview flowchart.

**Figure 6 sensors-24-02138-f006:**
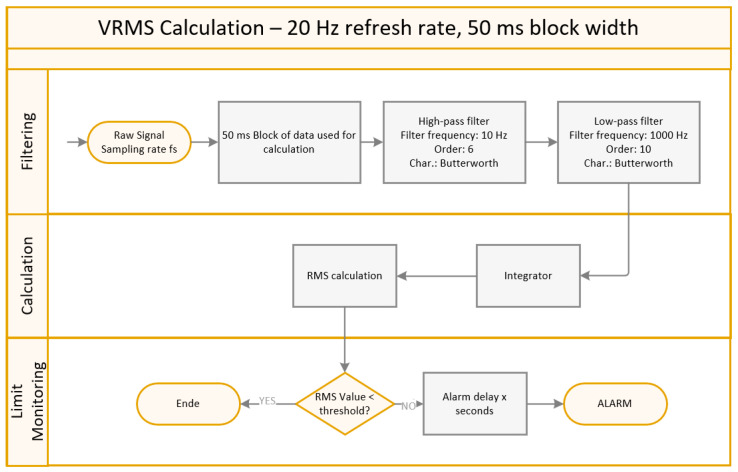
VRMS benchmarking algorithm.

**Figure 7 sensors-24-02138-f007:**
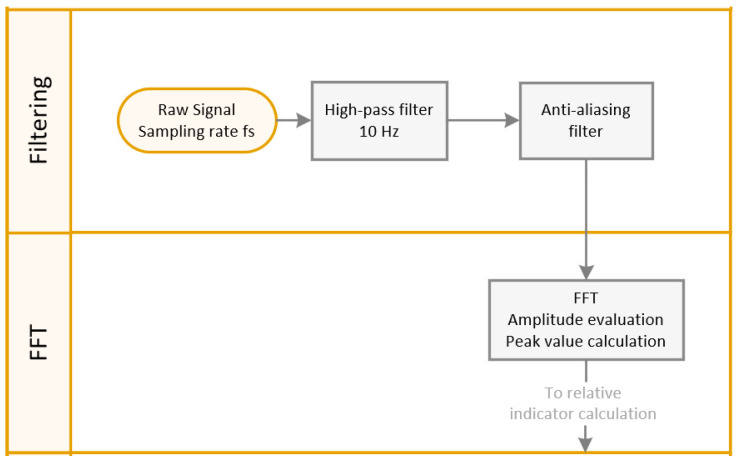
Signal conditioning and FFT block diagram.

**Figure 8 sensors-24-02138-f008:**
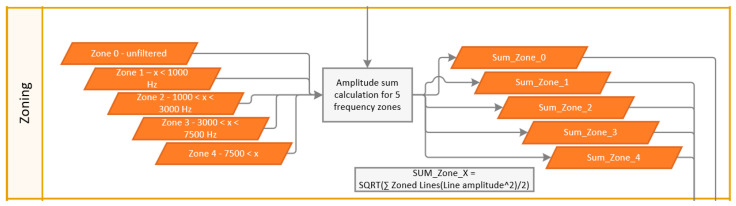
Zoning flowchart.

**Figure 9 sensors-24-02138-f009:**
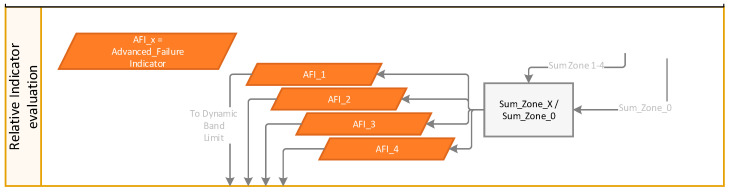
Relative indicator evaluation—AFI_x.

**Figure 10 sensors-24-02138-f010:**
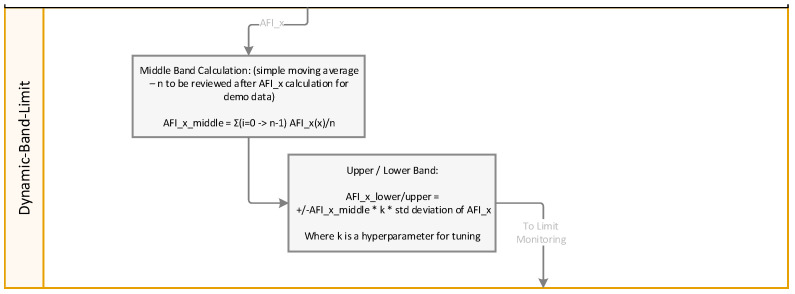
Dynamic band limit calculation.

**Figure 11 sensors-24-02138-f011:**
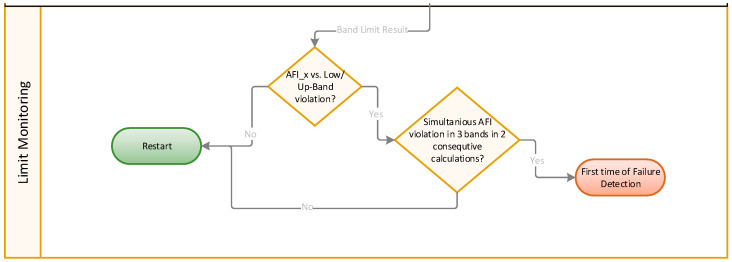
Limit monitoring—alarm decision and failure confirmation.

**Figure 12 sensors-24-02138-f012:**
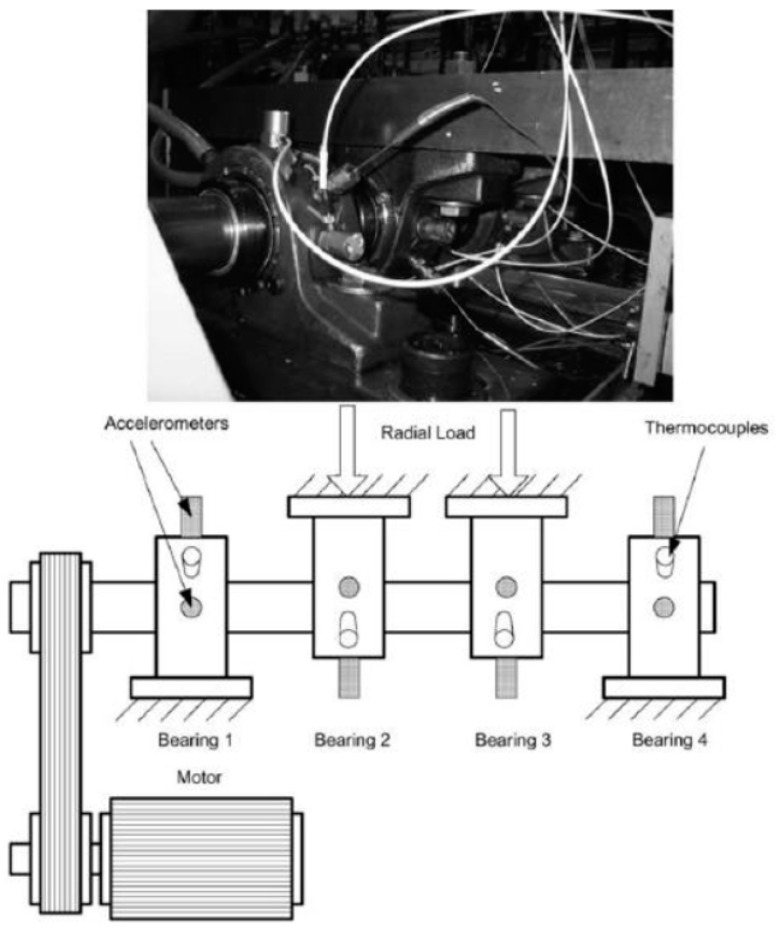
Layout of the NASA bearing dataset test setup. Source: https://www.kaggle.com/datasets/vinayak123tyagi/bearing-dataset, accessed on 15 February 2024.

**Figure 13 sensors-24-02138-f013:**
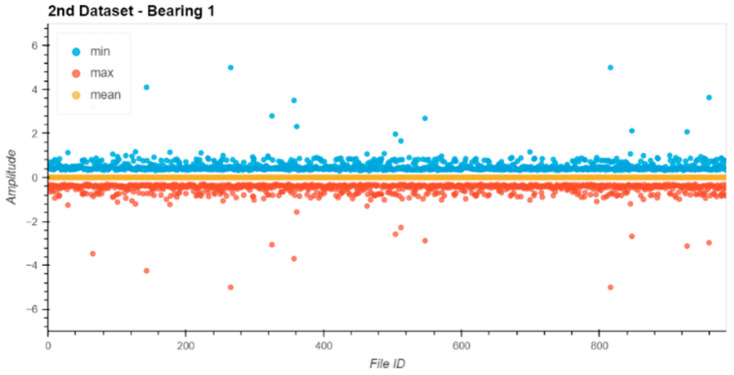
Dataset No. 2, Bearing No. 1: min/mean/max values for each file. VRMS in mm/s plotted vs. File ID.

**Figure 14 sensors-24-02138-f014:**
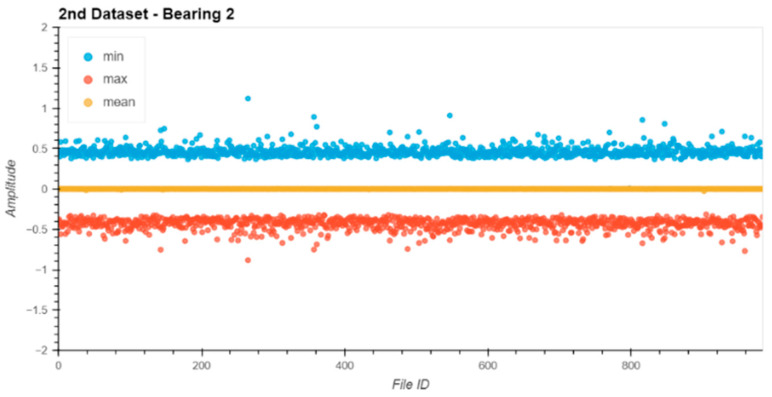
Dataset No. 2, Bearing No. 2: min/mean/max values for each file. VRMS in mm/s plotted vs. File ID.

**Figure 15 sensors-24-02138-f015:**
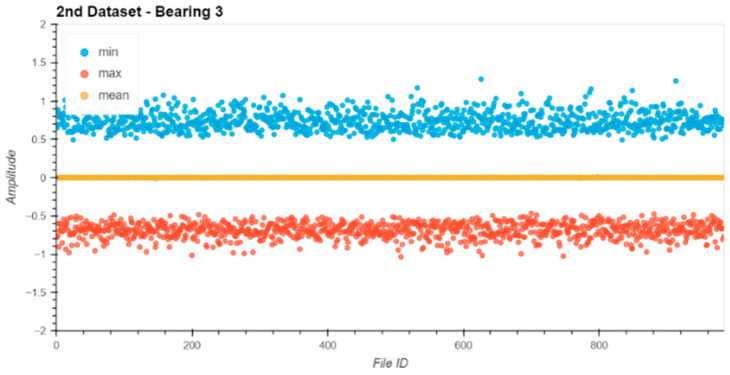
Dataset No. 2, Bearing No. 3: min/mean/max values for each file. VRMS in mm/s plotted vs. File ID.

**Figure 16 sensors-24-02138-f016:**
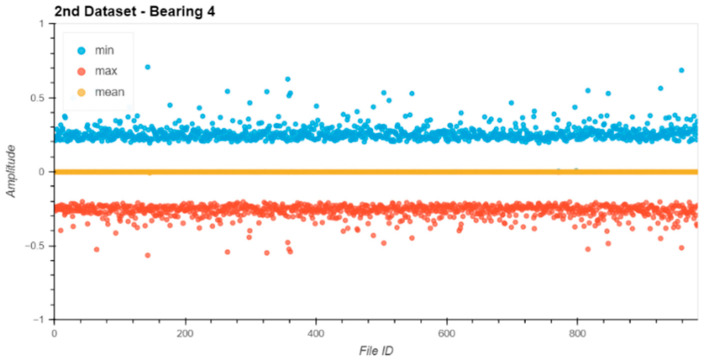
Dataset No. 2, Bearing No. 4: min/mean/max values for each file. VRMS in mm/s plotted vs. File ID.

**Figure 17 sensors-24-02138-f017:**
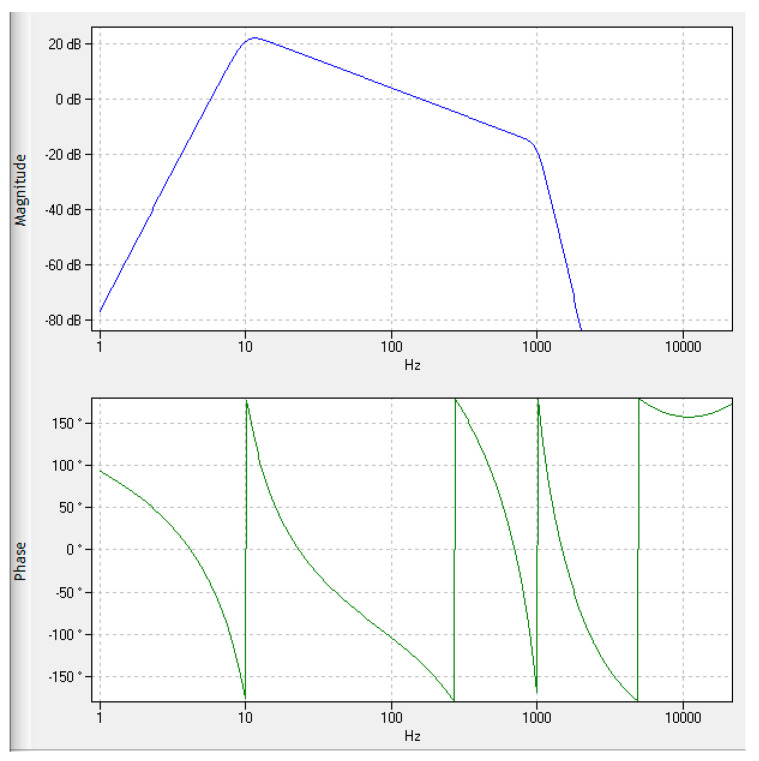
Bode plot of the VRMS calculation path.

**Figure 18 sensors-24-02138-f018:**
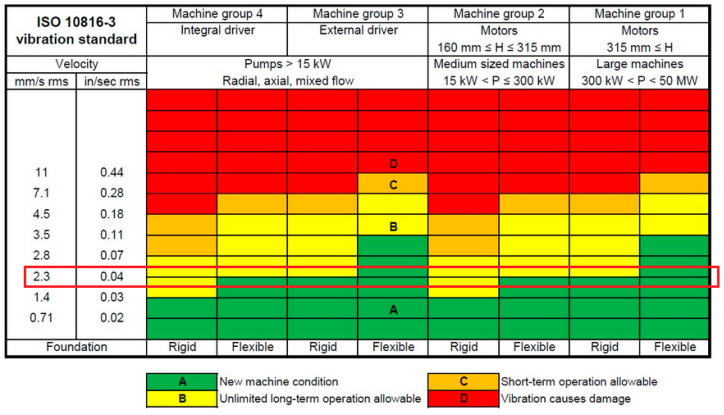
Permissible vibration velocity values based on ISO 10816-3. Source: https://www.cbmconnect.com/simplified-vibration-monitoring-iso-10816-3-guidelines/, accessed on 15 February 2024.

**Figure 19 sensors-24-02138-f019:**
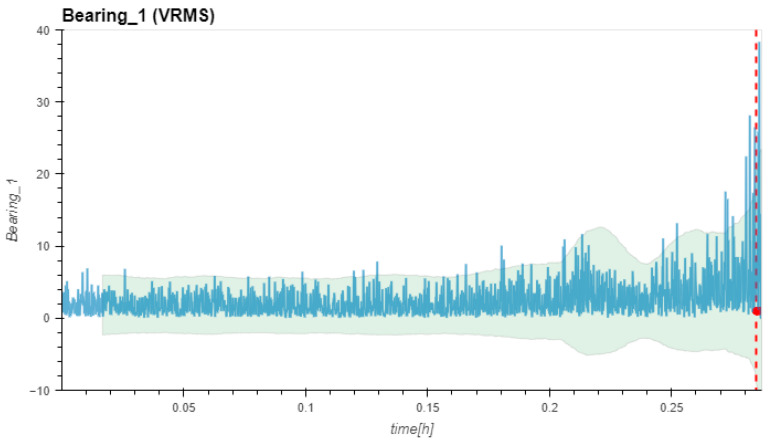
(V)RMS values in mm/s vs. measurement time in hours and FTVD prediction for Dataset No. 2.

**Figure 20 sensors-24-02138-f020:**
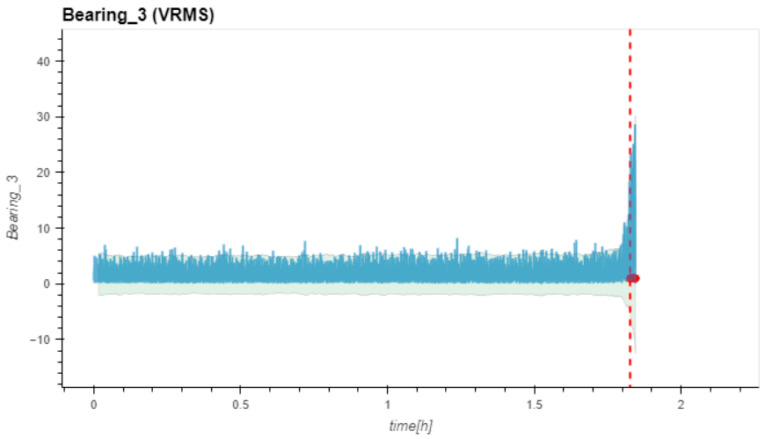
(V)RMS values vs. measurement time and FTVD prediction for Dataset No. 3. VRMS in mm/s plotted vs. measurement time in hours.

**Figure 21 sensors-24-02138-f021:**
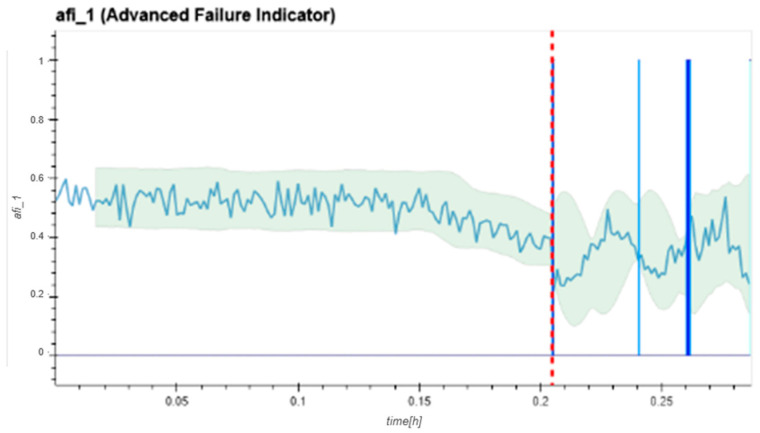
Frequency Zone 1 for Bearing No. 1 in Dataset 2. The FTFD is detected after 0.205 h of measurement time, corresponding to 120 h of runtime.

**Figure 30 sensors-24-02138-f030:**
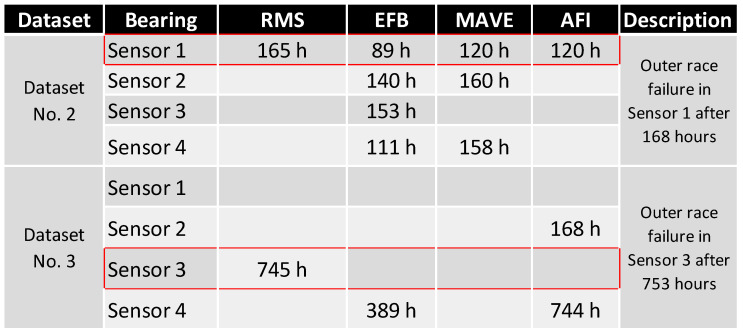
First time of failure detection by each of the compared methods.

**Figure 31 sensors-24-02138-f031:**
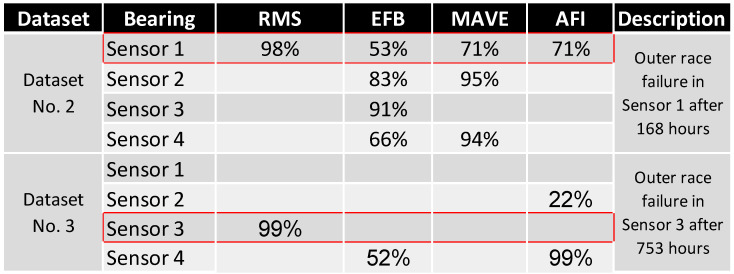
Relative first time of failure detection by each of the methods.

**Figure 32 sensors-24-02138-f032:**
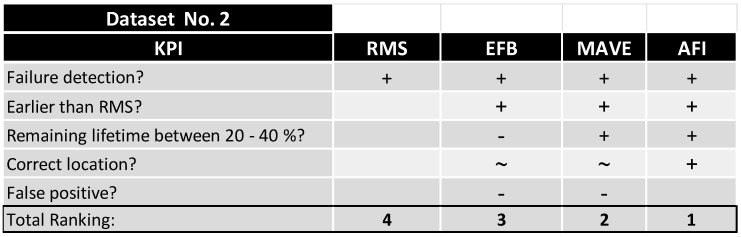
KPI evaluation and performance ranking, Dataset No. 2.

**Figure 33 sensors-24-02138-f033:**
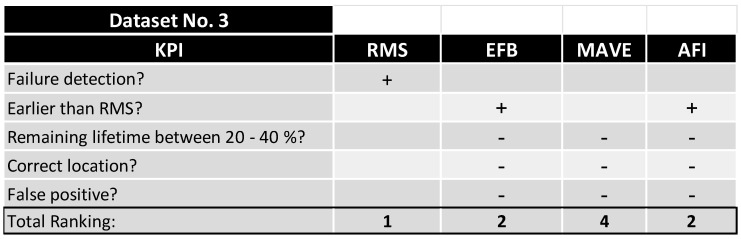
KPI evaluation and performance ranking, Dataset No. 3.

## Data Availability

The raw data supporting the conclusions of this article will be made available by the authors on request.

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
