# Peer review of "Evaluation of a Condition Monitoring Algorithm for Early Bearing Fault Detection"

_sensors, 2024, doi:10.3390/s24072138_

Round 1

Reviewer 1 Report

Comments and Suggestions for Authors

Manuscript ID: sensors-2877620

Title:
Evaluation of a condition monitoring algorithm for early bearing fault detection

Authors: Hannes Gruber * , Anna Fuchs , Michael Bader

General comments:

In conclusion, this article presents an algorithm, the Advanced Failure Indicator (AFI), designed for the early detection of roller bearing failures. Leveraging the Fast Fourier Transform (FFT) of wideband accelerometers, the AFI accurately calculates the spectral content of vibration signals emitted by roller bearings, aiding in predictive maintenance strategies. While the AFI demonstrates promise, it's important to acknowledge the robustness of other early detection algorithms for bearing failures, as evidenced in the literature, such as those showcased in Mechanical Systems and Signal Processing (MSSP).

Several points requiring attention have been identified for revision. Firstly, inconsistencies in referencing format are prevalent throughout the text, necessitating correction for clarity and accuracy. For instance, attention to detail is needed in ensuring proper citation placement and format adherence. Secondly, concerns regarding the presentation of testbed results, including issues with signal plot quality and absence of unit specification, highlight the need for improved data visualization practices to enhance analytical insights. Lastly, a thorough review of the reference list is warranted to rectify any inaccuracies or formatting discrepancies, ensuring adherence to established citation conventions.

Addressing these concerns will not only enhance the readability and credibility of the article but also contribute to its overall scholarly impact and effectiveness in disseminating valuable insights into early detection algorithms for bearing failures.

Some points to be corrected:

1)Throughout the text, there are numerous instances where references are incorrectly formatted. For example, on page 1, lines 42 and 44, among others. In line 44, the reference should be formatted as follows: "...USD in 2021 up to 64 bn USD in 2030 (2)."

2) On page 2, line 61, there's an issue with reference formatting:

testbed results in a loss of data quality and valuable development time (12) (13).

 It should be corrected to:

 testbed results in a loss of data quality and valuable development time (12,13).

These problems persist throughout the entire text.

3)The results section:

presents challenges in terms of signal plot quality and the absence of unit specification, making it difficult to analyse the data effectively.

4) In the reference section, several references (e.g., 7, 9, 10, 11) are not correctly formatted and need to be revised for accuracy and consistency.

Author Response

Dear Reviewer,

Thank you very much. Your response is highly appreciated. Please find the main revision points listed below. All changes can be traced using the MS Word markup function.

  1. Minor to moderate Language correction. Wording, phrasing, and punctuation are corrected, standardized, or improved.
  2. Revised format of references throughout the text
  3. The reference format in the reference section is standardized.
  4. Results section – absence of units. Added to figure description.
  5. The abstract section (Reviewers 2 & 4) has been simplified. The Focus of the research is highlighted, some complex phrasings are exchanged to enhance readability, and the last sentence is removed due to redundancy.
  6. VRMS Benchmarking algorithm explained in section 2.
  7. Unnecessary VRMS graphs in the results section were removed; relevant ones increased in size and quality.

An additional figure is added to explain the AFI graphs better. The nature of the four AFI graphs (e.g. new 21-28) was kept representing one total AFI state for each bearing.

  1. The Limitations chapter was reworked and, more detailed limitations were added, improvement points were highlighted.
  2. Figure 7 & 8 redone (clarity & size)
  3. Chapter 3 (Page 13) hyper-parameter values for evaluation explained
  4. Explanation for not detecting the error in dataset 3 added
  5. Innovation and benefits are highlighted in section 4.
  6.  

With kindest regards

The authors

In representation: Hannes Gruber

Reviewer 2 Report

Comments and Suggestions for Authors

1. The English writing quality of this article still has some room for improvement. The content is not very smooth and it is somewhat difficult to read.

2. The abstract part of the article is too redundant. It is suggested to simplify the content and highlight the key points.

3. The clarity of the pictures in this article needs to be improved, and it is suggested to adjust and improve.

4. Please introduce and supplement the limitations and improvement points of AFI in this article.

Comments on the Quality of English Language

The English writing quality of this article still has some room for improvement. The content is not very smooth and it is somewhat difficult to read.

Author Response

Dear Reviewer,

Thank you very much. Your response is highly appreciated. Please find the main revision points listed below. All changes can be traced using the MS Word markup function.

  1. Minor to moderate Language correction. Wording, phrasing, and punctuation are corrected, standardized, or improved.
  2. Revised format of references throughout the text
  3. The reference format in the reference section is standardized.
  4. Results section – absence of units. Added to figure description.
  5. The abstract section (Reviewers 2 & 4) has been simplified. The Focus of the research is highlighted, some complex phrasings are exchanged to enhance readability, and the last sentence is removed due to redundancy.
  6. VRMS Benchmarking algorithm explained in section 2.
  7. Unnecessary VRMS graphs in the results section were removed; relevant ones increased in size and quality.

An additional figure is added to explain the AFI graphs better. The nature of the four AFI graphs (e.g. new 21-28) was kept representing one total AFI state for each bearing.

  1. The Limitations chapter was reworked and, more detailed limitations were added, improvement points were highlighted.
  2. Figure 7 & 8 redone (clarity & size)
  3. Chapter 3 (Page 13) hyper-parameter values for evaluation explained
  4. Explanation for not detecting the error in dataset 3 added
  5. Innovation and benefits are highlighted in section 4.

With kindest regards

The authors

In representation: Hannes Gruber

Reviewer 3 Report

Comments and Suggestions for Authors

This paper uses the AFI algorithm to calculate the spectral content of the vibration signals emitted by 13 roller bearings using the Fast Fourier transform (FFT) of the wideband accelerometer for a comprehensive analysis of the mechanical health conditions focused on the early stages of failure. It has certain practical application value. But the article has the following problems:

1. In order to exclude the good data of  data itself, comparative experiments should be conducted to explore the authenticity of the results.

2. The introduction lacks the literature reference of the phased data set, resulting in the value of this paper derived from the insufficiency of previous work.

3. Please explain the benchmark algorithm, especially the VRMS algorithm mentioned many times in the article.

4. The innovation of the article lies in the stages, how to intuitively reflect the method proposed in the article, so as to make the significance of no stage exhaustive

To sum up, there are some innovations in this paper, but there are also many problems. Propose a revised retrial.

Comments on the Quality of English Language

The English expression of the article is clear, but the format and layout are a little confusing, and one English word cannot span two lines.

Author Response

(The authors gave the same response as above.)

Reviewer 4 Report

Comments and Suggestions for Authors

Your abstract presentation is too complex. It is recommended that the focus of the research in this paper be highlighted.

 The dataset you are using in the article is the NASA IMS dataset. So why do you say in the abstract that your proposed method is applicable to bearings running at high speeds? If so, you need to compare it with the low speed running ones under the same conditions.

In the introduction, you can introduce some scholars' research methods in the field.

 Figure 6 could be plotted more compactly. Please replace the curves with straight lines. The lower end of Figure 7 is not plotted in its entirety.

 Please specify what the hyperparameters are in section 3.2 for dataset 2 and dataset 3.

The legend for Figure 12-15 and 18-27 is fuzzy, please enlarge the legend to reimport the image.

Please explain why the AFI did not correctly detect the bearing 3 failure in section 3.2.

 Innovativeness is not highlighted in the concluding section of section IV.

Author Response

(The authors gave the same response as above.)

Round 2

Reviewer 1 Report

Comments and Suggestions for Authors

After the revision, I recommend accepting the article. 

It is suggested  to correct  correct line 152 from page 5 to:  

and a means the bearing's contact angle.

Author Response

Dear Reviewer,

Thank you very much for the second feedback.

The autocorrect error in line 152 has been reworked (missing “a” as a variable in the equation).

With kindest regards

The authors

In representation: Hannes Gruber

Reviewer 3 Report

Comments and Suggestions for Authors

The main revision points have been basically revised, the revised work description of the paper is complete, and the innovation points are clearly stated, but the following two modifications are still needed:

1.The reference format is incomplete, such as [5] [26], please complete it.

2.Adjust the clarity of Figure 3 if possible.

Author Response

Dear Reviewer,

Thank you very much for the second feedback.

According to your comments, the following changes have been made:

  • References have been corrected (Nr. 5, 7 and 26)
  • The options to improve the quality are limited since Figure 3 is from an external source. We improved the contrast, removed some minor errors in the figure, and therefore enhanced readability.

With kindest regards

The authors

In representation: Hannes Gruber